# Effect of Polyethylene-Grafted Maleic Anhydride on the Properties of Flue-Gas Desulfurized Gypsum/Epoxy Resin Composites

Fei Li, Hai Li, Juncheng Die, Yafeng Zhang, Yi Li, Mingyu Wang, Yang Cao, Kexi Zhang * and Jinchun Tu *

State Key Laboratory of Marine Resource Utilization in South China Sea, School of Materials Science and Engineering, Hainan University, Haikou 570228, China
* Correspondence: zhangkexi@hainanu.edu.cn (K.Z.); tujinchun@hainanu.edu.cn (J.T.)

**Abstract:** Waste resource utilization can save energy, reduce costs, and is one of the important means to protect the environment. Flue-gas desulphurized (FGD) gypsum is a common industrial by-product. These by-products are not only difficult to use, but also have serious impacts on the ecological environment. The conventional process of the industrial utilization of the calcium sulfate whisker pretreatment process leads to a low utilization rate of FGD gypsum, further increasing the consumption of resources and leading to secondary pollution. This study presents a method of preparing composites by adding FGD gypsum directly into epoxy resin with polyethylene-grafted maleic (PGM) anhydride as a compatibilizer of FGD gypsum/epoxy resin composites. Results showed weak tensile properties and impact properties of the composites when only FGD gypsum was added. When the amount of PGM added was 6 wt%, the tensile properties and impact properties of FGD gypsum/epoxy resin composites improved by 75% and 63%, and compared with the neat epoxy resin, the tensile properties and impact properties of FGD gypsum/epoxy resin composites, respectively, improved by 30% and 57%. Additionally, laser particle size analysis, X-ray diffraction (XRD) analysis, Fourier transform infrared spectroscopy (FT-IR), Scanning electron microscopy (SEM), energy-dispersive X-ray spectroscopy (EDS), a thermogravimetric analyzer (TGA), and a Differential scanning calorimeter (DSC) were used to examine the effects of PGM on the mechanical properties of FGD gypsum/epoxy resin composites and its mechanism of action. The recycling of FGD gypsum in resin materials has been extended in this study.

**Keywords:** composite material; epoxy resin; flue-gas desulfurized gypsum; polyethylene-grafted maleic anhydride

## 1. Introduction

Semidry desulfurization and wet desulfurization in a furnace are usually used in the circulating fluidized bed boiler in the power production of thermal power plants, and these two approaches lead to high calcium content in the fly ash and bottom slag of circulating fluidized beds and produce abundant flue-gas desulfurized (FGD) gypsum [1]. The gross production of FGD gypsum reached 550 million tons in China in 2016, and a voluminous amount was discarded [2]. FGD gypsum produced via flue-gas desulfurization in thermal power plants has attracted widespread attention because of the technical problems and widespread distribution associated with it, and relevant recycling research has been conducted. The resource utilization of FGD gypsum is a hot spot in the research of industrial solid waste at present. However, the utilization rate of FGD gypsum is low and the treatment of FGD gypsum is more complex. Moreover, FGD gypsum often requires calcination to be used, which not only causes additional energy consumption, but also becomes a burden on the environment.

FGD gypsum has been explored in several scientific studies. FGD gypsum application could significantly increase crop yield and improve soil quality [3]. FGD gypsum applica-

tion is a viable strategy for reclaiming sodic soils due to its positive effects on soil fertility and biochemistry and because it may contribute to soil ecosystem sustainability [4]. The FGD gypsum produced as an energy-plant waste byproduct has recently been advocated as a physiochemical remediation strategy for phosphorus (P) through sorptive removal [5]. In addition, FGD gypsum can also be used as a potential high-efficiency fluoride-removal material [6].

By calcinating FGD gypsum into FGD plastic and blending it with fly ash and ordinary Portland cement, the applicability of FGD gypsum has been extended to structural composite materials [7]. The inorganic cementitious materials prepared using circulating fluidized bed fly ash, carbide slag, and FGD gypsum reduces cost and carbon emissions relative to traditional mortar [8]. Li et al. [9] reported the magnesium oxysulfate cement (MOSC) could be effectively improved when FGD gypsum was mixed into the MOSC at the curing at 40 °C. FGD gypsum can also be used to prepare building composite blocks [10,11]. Additionally, Tabatabai et al. [12] reported enhanced mechanical properties of polyester resin by utilizing approximately 50% FGD gypsum content.

Epoxy resin (EP) is a typical thermosetting polymer that is widely used as structural adhesives, coatings, electronic packaging materials, and matrices of fiber-reinforced composites [13]. However, the toughness of epoxy resin is poor, and good interfacial compatibility with the FGD gypsum has not yet been established [14,15]. Epoxy resins are often toughened with highly elastic polymer materials or rigid nanomaterials. However, the compatibility between solid rubber and EP is usually poor, leading to severe phase separation and, consequently, a poor toughening effect [16]. Dittanet et al. [17] reported that the addition of silica nanoparticles did not have a significant effect on glass transition temperature (TG) or the yield stress of epoxy resin, i.e., the yield stress and TG remained constant regardless of silica nanoparticle size.

In order to improve the resource utilization rate of FGD gypsum, reduce energy consumption, and protect the environment, in this study, we propose preparing the FGD gypsum/epoxy resin composites via direct mixing of the FGD gypsum and epoxy resin to improve the resource utilization of FGD gypsum. However, direct incorporation of FGD gypsum into epoxy resin can significantly reduce the mechanical properties of FGD gypsum/epoxy resin composites. PGM not only has good workability and polyethylene properties, but also presents reactivity and strong polarity of polar molecules of maleic anhydride. PGM is an excellent compatible additive for composite-material preparation [18]. Polyethylene-grafted maleic (PGM) anhydride was added as the compatibilizer to improve the interfacial compatibility between the FGD gypsum and composite matrix and the mechanical properties of FGD gypsum/epoxy resin composites.

## 2. Experiment

### 2.1. Materials

The epoxy resin (E-51) was purchased from China Shenzhen Jitian Chemical Co., Ltd. The polyamide (PA) resin, as a curing agent, was supplied by China Yichun Junzheng New Material Co., Ltd. Untreated flue-gas desulfurized (FGD) gypsum, as a filler, was obtained from China Hainan Landao Environmental Protection Industry Co., Ltd. Polyethylene-grafted maleic (PGM) anhydride, the material used for the composites, was purchased from Dow Chemical Company (PGM: the particle size was 500 mesh, density: 0.95 g/cm$^3$, the melt flow index was 2.3 g/10 min, and the grafting rate was 1.1 wt%, Midland, MI, USA). The defoamer (XYS-6201, industrial-grade purity) was obtained from China Guangdong Yunfeng Technology Co., Ltd., Guangzhou, China.

### 2.2. Preparation Method

The composite ratios are shown in Table 1. In the first place, the FGD gypsum was put into an electrothermal blowing dry box (GZX-9070MBE, Shanghai Boxun Industrial Co., LTD, Shanghai, China) to dry for 24 h at 80 °C and screened with a 300-mesh screen. Then, the epoxy resin was mixed with the polyamide resin, and the booster electric mixer

(JJ-15, Changzhou Hua'ao Instrument Manufacturing Co., Ltd., Changzhou, China) was used for 5 min to make them evenly mixed at room temperature. Subsequently, the 300 mesh of FGD gypsum was evenly added into the epoxy resin/polyamide resin mixture, and the stirring continued for 15 min. Next, the PGM was added into the FGD gypsum/epoxy resin composites. The stirring time was 5 min. In the process of preparing the composites, the mechanical stirring made the composites produce bubbles. Therefore, a defoamer was added into the composites and stirred for 5 min to remove bubbles. Additionally, the composites were poured into the mold. Then, the composites were placed on a plate hot press (XH-406BEW-50-300, Dongguan Xihua Test Instrument Co., Ltd., Dongguan, China) for pressure curing at a temperature of 90 °C and pressure of 15 MPa for 40 min.

**Table 1.** Material proportions for the composites.

| Experimental Group | Epoxy Resin wt% | FGD Gypsum wt% | PGM wt% | PA Resin wt% | Defoamer wt% |
|---|---|---|---|---|---|
| Sample 1 | 50 | 0 | 0 | 40 | 10 |
| Sample 2 | 30 | 40 | 0 | 20 | 10 |
| Sample 3 | 27 | 40 | 3 | 20 | 10 |
| Sample 4 | 24 | 40 | 6 | 20 | 10 |
| Sample 5 | 21 | 40 | 9 | 20 | 10 |

*2.3. Characterization*

X-ray diffraction (XRD) analysis of the FGD gypsum used in this study was performed (DX-2700BH, Jiangsu Skyray Instrument Co., Ltd., Kunshan, China). Fourier transform infrared spectroscopy (FT-IR, Nicolet iS50 + continuum, Hangzhou Shiming Instrument Equipment Co., Ltd., Hangzhou, China) was used to analyze the surface groups of the PGM and samples. The microstructure of the samples was observed using scanning electron microscopy (SEM, Inspect F50, FEI, Thermo Fisher Scientific, Waltham, MA, USA). The distributions of FGD gypsum and PGM were observed with energy-dispersive X-ray spectroscopy (EDS). Thermogravimetric analysis (TGA) of particle samples was conducted using a thermogravimetric analyzer (Q600, American TA Company, Wilmington, DE, USA) under the following conditions: flowing air atmosphere; temperature from RT to 1000 °C; nitrogen as the protective gas; and heating rate, 10 °C/min. Differential scanning calorimeter (DSC) was used on a DISCOVER DSC250 at a heating rate of 10 °C/min under a nitrogen atmosphere; temperature from 20 °C to 250 °C. The tensile test was implemented using a universal tensile testing machine (AI-7000-SU2, GOTECH Testing Machine Co., Ltd., Taiwan, China) in accordance with GB/T 2567-2008, the Chinese national standard for tensile strength of epoxy resin. The impact test was conducted using a dial-type cantilever beam impact tester (XBL-22, Shenzhen Kai Strength Test Instrument Co., Ltd., Shenzhen, China) in accordance with GB/T 1843-2008, the Chinese national standard for the impact strength of epoxy resin.

Specific amount of the FGD gypsum was used to perform laser particle size analysis using wet-particle-size testing using Hydro 2000SM (Malvern Company, Malvern, Worcestershire, UK). X-ray diffraction (XRD) analysis of the FGD gypsum used in this study was performed (DX-2700BH, Jiangsu Skyray Instrument Co., Ltd., Kunshan, China). Fourier transform infrared spectroscopy (FT-IR, Nicolet iS50 + continuum, Hangzhou Shiming Instrument Equipment Co., Ltd., Hangzhou, China) was used to analyze the surface groups of the PGM and samples. The microstructure of the samples was observed with scanning electron microscopy (SEM, Inspect F50, FEI, Waltham, MA, USA). The distributions of FGD gypsum and PGM were observed with energy-dispersive X-ray spectroscopy (EDS). Thermogravimetric analysis (TGA) of particle samples was conducted using a thermogravimetric analyzer (Q600, American TA Company, Wilmington, DE, USA) under the following conditions: flowing air atmosphere; temperature from RT to 1000 °C; nitrogen as the protective gas; and heating rate, 10 °C/min. Differential scanning calorimeter (DSC) was used on a DISCOVER DSC250 (American TA Company, Wilmington, DE, USA) at a heating rate of 10 °C/min under a nitrogen atmosphere; temperature from 20 °C to 250 °C.

The tensile test was implemented using a universal tensile testing machine (AI-7000-SU2, GOTECH Testing Machine Co., Ltd., Taiwan, China) in accordance with GB/T 2567-2008, the Chinese national standard for tensile strength of epoxy resin. In accordance with GB/T 2567-2008, the loading speed of the tensile strength of the samples was set to 2 mm/min. The impact test was conducted using a dial-type cantilever beam impact tester (XBL-22, Shenzhen Kai Strength Test Instrument Co., Ltd., Shenzhen, China) in accordance with GB/T 1843-2008, the Chinese national standard for the impact strength of epoxy resin.

### 3. Results and Discussion

As shown in Figure 1, flue-gas desulfurized (FGD) gypsum has a small particle size; 3–60 μm is the main distribution range. Figure 2 shows the XRD analysis result for FGD gypsum. The composition of FGD gypsum is relatively pure, with calcium sulfate hemihydrate as the main component.

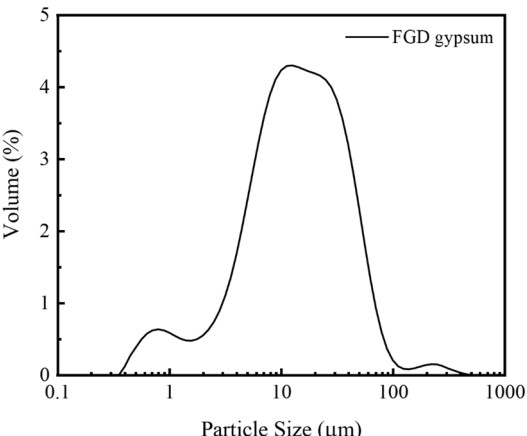

**Figure 1.** Particle size curve of FGD gypsum.

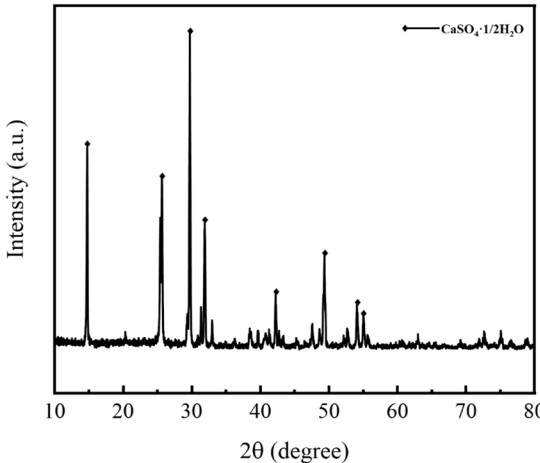

**Figure 2.** XRD analysis result for FGD gypsum.

Figure 3a shows the FT-IR spectra of PGM. The peaks at 2922 cm$^{-1}$ and 2846 cm$^{-1}$ represent methylene C–H stretching vibration contained in maleic anhydride grafted polyethylene powder, the peaks at 1460 cm$^{-1}$ are the methyl C–H stretching vibration, and the peaks at 720 cm$^{-1}$ are the C–H out-of-plane bending vibration of methyl.

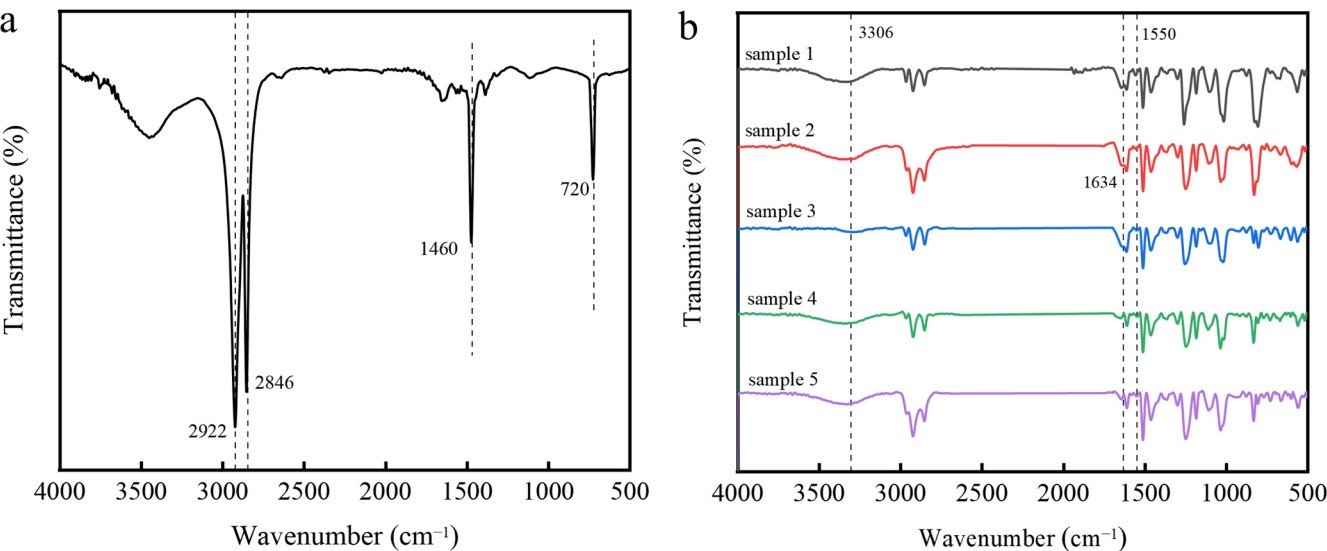

**Figure 3.** FT-IR spectra of PGM (**a**) and samples 1 to 5 (**b**).

Figure 3b shows the FT-IR spectra of samples 1 to 5. The peak at 3306 cm$^{-1}$ represents the –OH stretching vibration of the free water in each substance. The peaks at 1550 cm$^{-1}$ in samples 1 to 5 represent C=N stretching vibrations, which weaken after the addition of FGD gypsum (sample 2) and remain unchanged after the addition of PGM (samples 3 to 5). This is due to FGD hindering the crosslinking reaction between epoxy resin and PA [19].

Peaks at 1634 cm$^{-1}$ can be observed for samples 1 to 5. Additionally, the peaks at 1634 cm$^{-1}$ represent the vibration absorption peak of –C=O. These peaks remained constant with the addition of FGD gypsum (samples 1 and 2). Then, these peaks gradually weakened with the addition of PGM (samples 3 to 4), but slightly strengthened with the excessive addition of PGM (sample 5). The carbonyl group was a polar group. It only adsorbed with Ca$^+$ after adding FGD gypsum (sample 2), and did not react with FGD gypsum. Therefore, the peak at 1634 cm$^{-1}$ did not change after adding FGD gypsum (sample 2). After the addition of PGM, the carbonyl group in PGM and the carbonyl group in epoxy resin adsorbed each other, and the peak (sample 4) was weakened. When excess PGM was added, the carbonyl group increased. Therefore, the peak (sample 5) was slightly strengthened. In this study, the reactive anhydride group reacted with the amino group at the end of the PA molecule to form an amide bond first and then an imide bond through a closed loop to generate a PE-g-PA graft copolymer [20–22].

The microstructure of the samples was revealed in the SEM images (Figure 4). The fracture of sample 1 was smooth and flat, and sample 1 presented a relatively smooth crack pattern of the cracked surface, which was a typical brittle failure mode. The fracture surfaces became rougher with the addition of FGD gypsum and PGM (samples 2 to 5). However, the fracture surface of sample 2, with only the FGD gypsum added, was much smoother than that of samples 3 to 5, with both FGD gypsum and PGM added. Notably, a handful of transverse fracture (red circle) were observed in the crack initiation zone of sample 4, compared with samples 2, 3, and 5. This indicated that the stress of sample 4 was relatively dispersed during fracture [23].

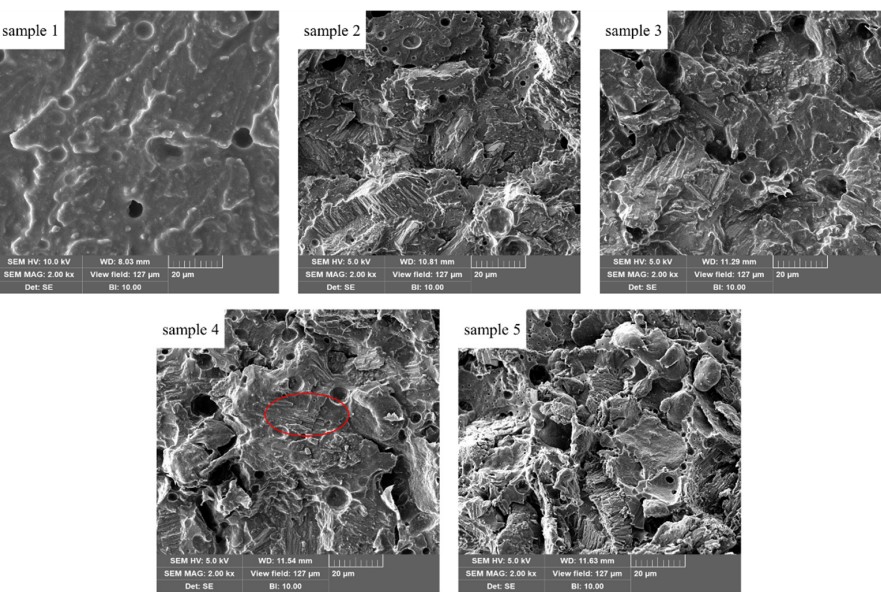

**Figure 4.** SEM images of samples 1 to 5.

When subjected to stress, the blend of calcium sulfate hemihydrate and PGM acted as stress concentrators; this phenomenon was followed by the formation of a triaxial stress field surrounding the blend of calcium sulfate hemihydrate and PGM, which further led to the formation of cavities via blend debonding or blend cavitation [24]. These cavities relieved the stress in front of the crack tip, which allowed the composite cavities to grow and enhance the mechanical properties of the composites [25]. This process altered the stress state of the surrounding matrix from a plane-strain state to a plane-stress state and caused plastic deformation and the yielding and crazing of the matrix; consequently, the mechanical properties of composites were improved [26].

Figure 5 shows the EDS images of samples 1 to 5. The distributions of FGD gypsum and PGM were observed with EDS. With the addition of PGM, the distribution of FGD gypsum at the fracture interface gradually became uniform (samples 3 and 4). This phenomenon indicates the ability of PGM to improve the interface compatibility between FGD gypsum and the base material. However, FGD gypsum also aggregated from sample 5. The excessive addition of PGM likely caused the aggregation, which led to the poor dispersion of FGD gypsum.

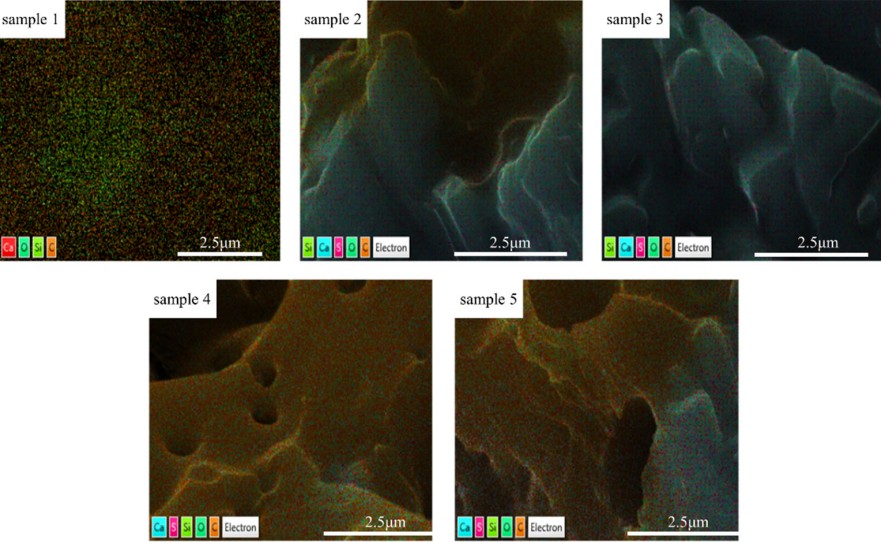

**Figure 5.** EDS images of samples 1 to 5.

Figure 6a shows the TGA curves of samples 1 to 5. Figure 6b shows the derivative TGA curves corresponding to Figure 6a, indicating the rate of temperature change in the thermal weight loss of the samples. In the initial temperature range to 500 °C, the weight loss of sample 1 accelerated significantly, and the weight loss was the most obvious, with a weight loss of 97.01%. Samples 2 to 5 were dehydrated at 230 °C and partially semi-hydrated calcium sulfate to anhydrite III, and at 500 °C partially dehydrated anhydrite III to anhydrite II. The other part of the semi-hydrated calcium sulfate, at 850 °C, directly dehydrated into type II anhydrite. Therefore, in the range of 780 °C to 850 °C, the weight loss of samples 2 to 5 accelerated significantly. The above information can well-illustrate the adsorption of $Ca^+$ with epoxy resin in the matrix material [27,28].

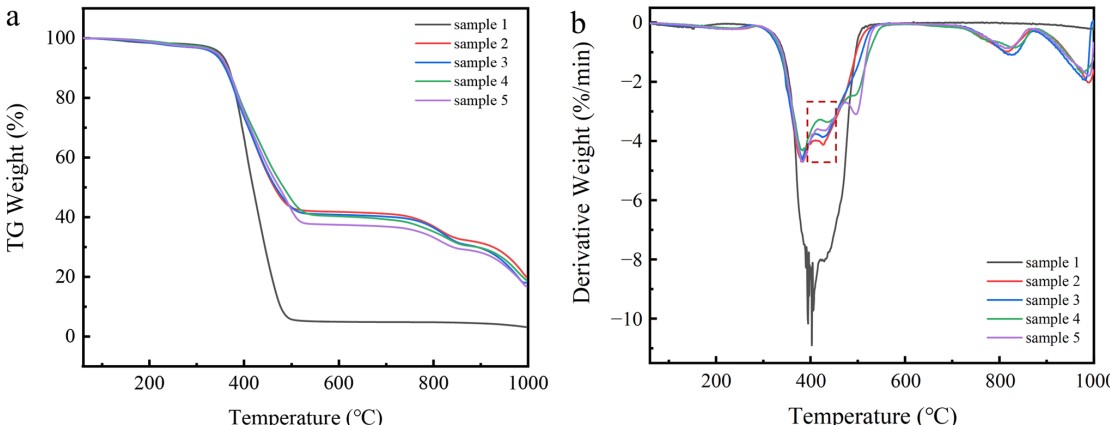

**Figure 6.** TGA curve (**a**) and DTG curve (**b**) of samples 1 to 5.

As can be seen from Figure 6b, in the initial temperature range to 500 °C, after the addition of FGD gypsum (sample 2), the decomposition rate of the composite was slowed down, and the thermal stability of the composite was improved. After adding PGM (samples 3 to 5), the thermal stability of the composite material was further improved, and the decomposition rate of the composite material was further slowed down. In particular, the thermal stability of sample 4 was the best among the five samples.

The curing behaviors of the composites were investigated using a DSC. As can be seen in Figure 7, the addition of FGD gypsum (sample 2) had very little impact on the onset, peak, and final temperatures of the curing reactions. When adding PGM to the composites, the endothermic peak, which was the melting heat absorption peak of polyethylene, of PGM can be observed (samples 3 to 5). At the same time, the endothermic peak continued to rise with the increase in PGM addition.

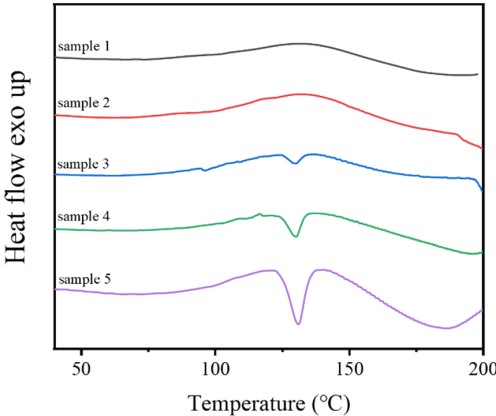

**Figure 7.** DSC thermograms of samples 1 to 5.

Figure 8 and Table 2 show the effects of PGM on the tensile strength and impact strength of the composite material. As shown in Figure 8a, the addition of FGD gypsum led to a decrease in the tensile strength of the composites. The tensile strength of sample 2 was nearly 25% lower than that of sample 1. For sample 4, when PGM was added to the composites as the compatibilizer, its tensile strength significantly improved by approximately 30% compared with that of sample 1. The impact strength of the composites is shown in Figure 8b. The impact strength of sample 2 was approximately 4% lower than that of sample 1. For sample 4, when PGM was added to the composites as the compatibilizer, its impact strength increased by approximately 57% compared with that of sample 1.

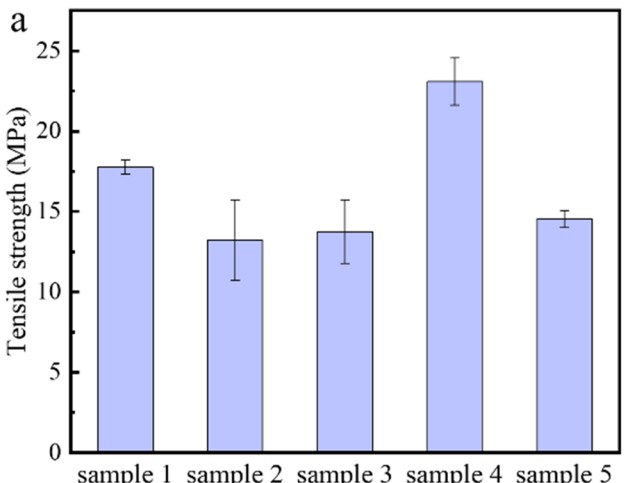 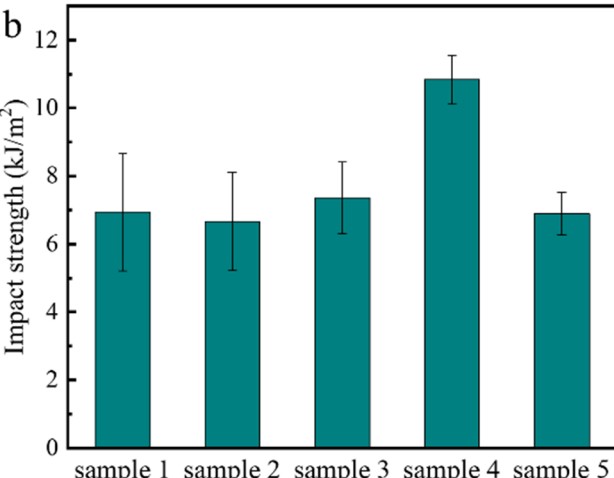

**Figure 8.** Tensile strength (**a**) and impact strength (**b**) of samples 1 to 5.

**Table 2.** Mechanical properties of FGD gypsum/epoxy resin composites with different PGM contents.

| Experimental Group | Sample 1 | Sample 2 | Sample 3 | Sample 4 | Sample 5 |
|---|---|---|---|---|---|
| Tensile strength (MPa) | $17.77 \pm 0.43$ | $13.21 \pm 2.49$ | $13.75 \pm 1.97$ | $23.09 \pm 1.47$ | $14.52 \pm 0.51$ |
| Impact strength (kJ/m$^2$) | $6.94 \pm 1.73$ | $6.66 \pm 1.44$ | $7.36 \pm 1.04$ | $10.83 \pm 0.72$ | $6.89 \pm 0.62$ |

Thus, the addition of PGM improved the interface compatibility between FGD gypsum and the matrix material (Figure 9). The addition of PGM (sample 4) not only solved the problem that the mechanical properties of the composites decreased due to the addition of FGD gypsum, but also improved the mechanical properties of the composites after the addition of PGM compared with neat epoxy resin (sample 1). In addition, the maleic anhydride group caused the epoxy group to open the ring and react with the epoxy resin to form a strengthened three-dimensional structure [29]. An esterification reaction occurred between the maleic anhydride group and secondary amide group, and a chemical link between the PGM and matrix material was formed [30]. This indicated that PGM can promote the crosslinking curing reaction of the composites, because when PGM was added to the composites, the anhydride group and epoxy group formed crosslinking propagation through ester and ether bonds [31,32]. The chemical link promoted the linkage between the FGD gypsum and the base material.

**Reaction process of flue gas desulfurized gypsum/epoxy resin composites**

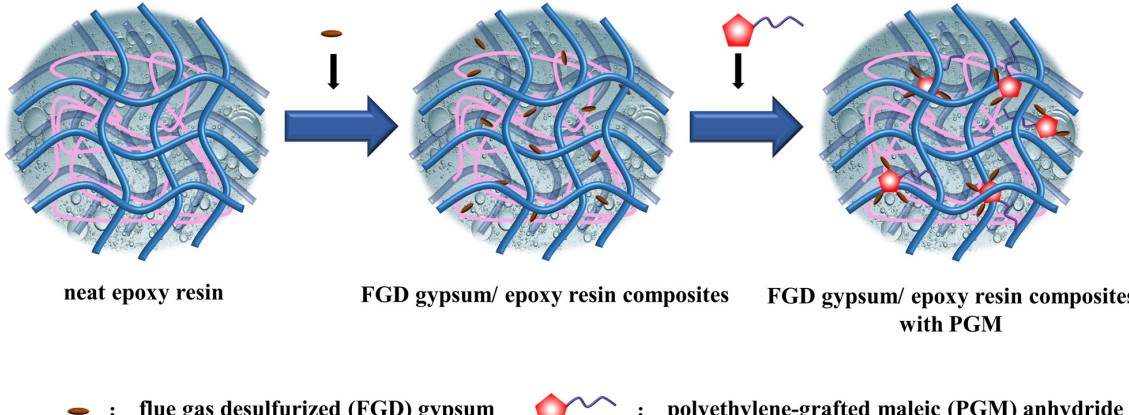

neat epoxy resin    FGD gypsum/ epoxy resin composites    FGD gypsum/ epoxy resin composites with PGM

⬮ :  **flue gas desulfurized (FGD) gypsum**         ⬠〜 :  **polyethylene-grafted maleic (PGM) anhydride**

〜 :  **polyamide (PA) resin**

**Figure 9.** Analysis model of the influence of polyethylene-grafted maleic anhydride on flue-gas desulfurized gypsum/epoxy resin composites.

## 4. Conclusions

Due to the inherent high cross-linked structure of epoxy resin, the brittleness is great. When flue-gas desulfurized (FGD) gypsum was mixed into epoxy resin to prepare FGD gypsum/epoxy resin composites, the mechanical properties of FGD gypsum/epoxy resin composites will be reduced. FGD gypsum/epoxy resin composites with good mechanical properties were prepared in this study by adding PGM. The addition of PGM to FGD gypsum/epoxy resin composites improved the interface compatibility between the FGD gypsum and matrix material. However, the addition of excessive PGM to FGD gypsum/epoxy resin composites will lead to the agglomeration of FGD gypsum, which will make its dispersibility worse in the matrix material. The results indicate that 6 wt% was the best amount of PGM added. The tensile strength and impact strength of FGD gypsum/epoxy resin composites enhanced by 75% and 63%. In addition, the tensile strength and impact strength of FGD gypsum/epoxy resin composites were increased by approximately 30% and 57%, respectively, compared with those of the neat epoxy resin. This study expands the engineering application of solid waste reuse of FGD gypsum and provides a wide range of possibilities for the further efficient utilization of FGD gypsum.

**Author Contributions:** Conceptualization, F.L.; methodology, F.L.; formal analysis, K.Z.; investigation, H.L., J.D., Y.Z., Y.L., M.W. and Y.C.; re-sources, K.Z. and J.T.; writing—original draft preparation, F.L.; writing—review and editing, K.Z. and J.T. All authors have read and agreed to the published version of the manuscript.

**Funding:** We gratefully acknowledge the technology project of comprehensive utilization and development of flue gas desulfurized gypsum (HD-KYH-2019109) and the study on strengthening and application of green building materials in desalination sea sand concrete engineering (ZDYF2021GXJS212).

**Institutional Review Board Statement:** Not applicable.

**Informed Consent Statement:** Not applicable.

**Data Availability Statement:** Data openly available in a public repository.

**Acknowledgments:** We acknowledge Mingwang Liu (China Hainan Landao Environmental Protection Industry Co., Ltd.) for providing device support.

**Conflicts of Interest:** The authors declare no conflict of interest.

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
