# Peer review of "Effect of Polyethylene-Grafted Maleic Anhydride on the Properties of Flue-Gas Desulfurized Gypsum/Epoxy Resin Composites"

_coatings, doi:10.3390/coatings13071291_

Round 1

Reviewer 1 Report

Authors are advised to include more details about

tensile strength and impact strength of composites measurements along with the equipment details in experimental section and results and discussion part, that would help the other  to reproduce their results. 

also include calibration of the those equipment.

Reviewer 2 Report

The paper deals on the fabrication and characterization of composite materials constituted by gypsum coming from desulfurizing process and epoxy resin. 

First of the state of the art is not clearly described in the introduction part and it is my opinion there are a plenty of more reasonable and suitable ways to dispose of such an industrial residue rather than to make a poorly rinforced and non recyclable composite. More importantly, the topic of this paper looks far from the aims and scopes of "Coatings". Therefore , in my opinion, the manuscript sould be redirected for publication in a different Journal.

However, there are some technical point that had to be corrected anyway: 

introduction paragraph: acronims EP and TG have been introduced without explanation of their meaning

Figure 1 displays a size distribution curve, however no indication about of this instrument/technique used is provided in paragraph 2.3.

Figure 1 b . XRD spectra are almost unreadable. The main peaks are assigned to gypsum but what about the others? Do these crystalline phases play a role in the mechanical properties?

Figure 3 SEM images and EDS mapping almost unreadable. Please enlarge them.

Reviewer 3 Report

The proportions of materials for the composites are given in Table 1. It is thought that the ratios are given in percentages. However, the % symbol is not written in the table. Moreover, the total of the example ratios does not add up to 100. A detailed explanation is needed.

The abstract needs to be updated to include the results of the study.

The results of the study should be interpreted by comparing them with the literature.

The conclusion section is very brief. It needs to be expanded.

Round 2

Reviewer 2 Report

The Authors succesfully replied to my concerns. The paper can be saccepted for publication.